# Development and Transportation Pathway Evaluation of Liposomes with Bile Acids for Enhancing the Blood-Brain Barrier Penetration of Methotrexate

**DOI:** 10.3390/pharmaceutics17020269

**Published:** 2025-02-17

**Authors:** Natthan Charernsriwilaiwat, Rattanan Thaitrong, Samarwadee Plianwong, Praneet Opanasopit, Pucharee Songprakhon, Thirapit Subongkot

**Affiliations:** 1Department of Pharmaceutical Technology, Faculty of Pharmaceutical Sciences, Burapha University, Chonburi 20131, Thailand; natthan@go.buu.ac.th (N.C.); 65910175@go.buu.ac.th (R.T.); 2Department of Pharmaceutical Chemistry, Faculty of Pharmaceutical Sciences, Burapha University, Chonburi 20131, Thailand; samarwadee.pl@go.buu.ac.th; 3Department of Industrial Pharmacy, Faculty of Pharmacy, Silpakorn University, Nakhon Pathom 73000, Thailand; opanasopit_p@su.ac.th; 4Division of Molecular Medicine, Research Department, Faculty of Medicine Siriraj Hospital, Mahidol University, Bangkok 10700, Thailand; pucharee.son@mahidol.ac.th

**Keywords:** methotrexate, liposomes, bile acids, blood-brain barrier penetration, transportation pathways

## Abstract

**Background/Objectives**: The purpose of this study was to create bile acid-containing liposomes to improve methotrexate blood-brain barrier penetration and to assess the liposome transportation mechanism across the blood–brain barrier. **Methods**: The improvement of liposome penetration was investigated utilizing human brain microvascular endothelial cells in an in vitro blood-brain barrier model. Using confocal laser scanning microscopy (CLSM) and flow cytometry, liposomes were labeled with fluorescent phospholipids to facilitate their passage across the blood–brain barrier. **Results**: The produced liposomes with bile acid exhibited a negative surface charge and an average particle size of between 30 and 148 nm. According to an in vitro blood-brain barrier penetration study, the methotrexate penetration was increased by liposomes containing 1% glycocholic acid but not by liposomes containing taurocholic acid. For transport pathway evaluation across the blood-brain barrier of these liposomes, CLSM revealed that fluorescent liposomes were present inside cells treated with specific endocytosis inhibitors, indicating that the cellular internalization of the particles was not involved in endocytosis. **Conclusions**: Liposomes supplemented with 1% glycocholic acid could enhance the penetration of methotrexate across the blood-brain barrier, while taurocholic acid could not. The transport of liposomes with 1% glycocholic acid across the blood-brain barrier occurs via the transcellular pathway through which it penetrates cells. In contrast, the paracellular pathway was a minor pathway.

## 1. Introduction

Acute lymphoblastic leukemia (ALL) is a hematologic malignancy that affects both children and adults. It arises from the unchecked proliferation and aberrant differentiation of lymphoid progenitor cells in the bone marrow [1]. The presence of leukemic blast cells in the central nervous system (CNS) is associated with increased risks of morbidity and mortality [2,3]. To prevent CNS involvement in ALL, current treatment approaches include radiation therapy, systemic chemotherapy, and intrathecal drug administration [4,5]. However, cranial irradiation causes serious side effects, such as endocrinopathy and neurotoxicity. Cranial irradiation is currently replaced by systemic and intrathecal therapy. For systemic therapy, high-dose methotrexate (5–8 g/m^2^) was used for prophylaxis in regard to CNS involvement [6]. The use of high-dose methotrexate is limited due to the increase in widespread toxicity and difficulty in maintaining the drug concentration in cerebrospinal fluid (CSF). Thus, intrathecal methotrexate has been widely used because of its ability to prolong drug concentrations in CSF. Intrathecal chemotherapy involves the administration of drugs into CSF via lumbar puncture or an Ommaya reservoir injection to avoid the problem of drug penetration through the blood-brain barrier [7]. However, intrathecal chemotherapy has adverse effects, such as back pain, headache, and paresthesia [8]. Furthermore, intrathecal chemotherapy must be administered by specialized physicians such as medical oncologists and hematologists. A drug delivery system is needed that delivers a sufficient therapeutic amount of methotrexate into the CNS via intravenous injection, which is more convenient than the intrathecal route.

Drug delivery between the blood and CNS is strongly restricted by the blood-brain barrier, which is the complex structure of cerebral microvessels [9]. The blood-brain barrier system consists of cerebral endothelial cells, which are connected by tight junctions and surrounded separately by astrocytes, pericytes, microglia, and neurons [10]. Because it is a semipermeable membrane of the blood-brain barrier, lipophilic molecules can permeate the CNS more than hydrophilic molecules.

Methotrexate is an antimetabolite cytotoxic drug that inhibits dihydrofolate reductase. It is used in chemotherapy and the treatment of autoimmune diseases. Methotrexate has a molecular weight and log partition coefficient (log P) of 454.56 g/mol and −1.85, respectively [11]. Due to its low log P value, methotrexate exhibits poor permeability across the blood-brain barrier.

Liposomes are spherical nanoparticles mainly composed of phospholipids and cholesterol. Structurally, they contain mono- or multilipid bilayers that can entrap both hydrophilic and lipophilic molecules. Several publications have reported the delivery of liposomes with chemotherapy drugs, such as irinotecan [12], topotecan [13], and paclitaxel [14], across the blood-brain barrier.

Bile acids are natural biological surfactants synthesized from cholesterol in the liver. Bile acids conjugated with amino acids, namely, glycine and taurine, are glycocholic acid and taurocholic acid, respectively. Conjugated bile acids are stored in the gallbladder and secreted into the intestine to emulsify lipids for absorption [15]. Due to their advantages in biocompatibility, conjugated bile acids are used in pharmaceutical formulations. In the pharmaceutical industry, glycocholic acid is used to solubilize vitamin K for intravenous injection (Konakion®MM) [16]. However, the effect of conjugated bile acids on the blood-brain barrier penetration of drugs has never been evaluated before. Therefore, this study investigated the ability of liposomes containing glycocholic acid or taurocholic acid to enhance the penetration of the blood-brain barrier by methotrexate using an in vitro blood-brain barrier model. There were publications reporting the effect of particle size on the blood-brain barrier penetration enhancement of silica nanoparticles [17] and pegylated gold nanoparticles [18]. However, the effect of liposome size on blood-brain barrier penetration enhancement has not been examined. This study also initiated the influence of liposome size on blood-brain barrier penetration enhancement. This study also investigated the transport pathway of liposomes with bile acids across the blood-brain barrier.

## 2. Materials and Methods

### 2.1. Materials

Methotrexate was purchased from Tokyo Chemical Industry Co., Ltd., Tokyo, Japan. Phospholipids (Lipoid S100) were donated by Lipoid GmbH, Ludwigshafen, Germany. Glycocholic acid, taurocholic acid, chlorpromazine hydrochloride, genistein, filipin, amiloride hydrochloride, cholesterol, 4′,6-Diamidino-2-phenylindole dihydrochloride, 2-(4-Amidinophenyl)-6-indolecarbamidine dihydrochloride (DAPI dihydrochloride), and endothelial cell growth supplement (ECGS) from bovine neural tissue were purchased from Sigma Aldrich, St. Louis, MO, USA. Alexa Fluor™ 488 phalloidin and Lissamine™ rhodamine B 1,2-dihexadecanoyl-sn-glycero-3-phosphoethanolamine trimethylammonium salt (rhodamine DHPE) were purchased from Invitrogen, Carlsbad, CA, USA. All other reagents were of analytical grade and were commercially available.

### 2.2. Preparation of Phosphate Buffer (pH 8), Methotrexate Stock Solution, and Liposomes

#### 2.2.1. Preparation of Phosphate Buffer at pH 8

Phosphate buffer at pH 8.0 was prepared by first weighing 0.81 and 13.35 g of KH_2_PO_4_ and Na_2_HPO_4_, respectively. Then, ultrapure water was added to dissolve KH_2_PO_4_ and Na_2_HPO_4_, and the volume was adjusted to 1 L. Phosphoric acid (85 %) or 1 M NaOH was added to adjust the pH to 8.0.

#### 2.2.2. Preparation of Methotrexate Stock Solution

To prepare a methotrexate stock solution (17 mg/mL), 85 mg of methotrexate was weighed into a volumetric flask. Then, 4 mL of phosphate buffer (pH 8) was added, and the mixture was sonicated until completely dissolved. Phosphate buffer (pH 8) was added to adjust the volume to 5 mL.

#### 2.2.3. Preparation of Methotrexate Solution and Methotrexate-Loaded Liposomes

Five hundred microliters of the methotrexate stock solution was pipetted into a volumetric flask to create the methotrexate solution. Then, the volume was adjusted to 5 mL by adding phosphate buffer (pH 8). Table 1 displays the components of various liposomal formulations loaded with methotrexate. Lipoid S100 (0.773 g) was weighed in a volumetric flask to create a phospholipid stock solution and adjusted to 5 mL by chloroform–methanol (2:1 *v*/*v*). Cholesterol was weighted at 0.0773 g into a volumetric flask and adjusted to 10 mL with chloroform–methanol (2:1 *v*/*v*) to produce the cholesterol stock solution. The thin film method was used to create liposomes loaded with methotrexate. Five hundred microliters of cholesterol and 250 µL of phospholipid stock solution were pipetted into a test tube to create a thin film layer. A mild stream of nitrogen gas was then used to evaporate the combination of phospholipid and cholesterol solution. Overnight, the test tube with thin film was placed in a desiccator. Methotrexate-loaded liposomes were prepared by pipetting 0.5 mL of methotrexate stock solution (17 mg/mL) and 4.5 mL of phosphate buffer (pH 8) into a dried lipid film test tube. The hydrated thin film was vortexed vigorously with a vortex mixer (Vortex-Genie 2, Scientific Industries, Inc., Bohemia, NY, USA) until the film detached from the test tube wall. The obtained liposomes were transferred into a glass vial, which was placed in an ice bath to reduce the particle size using a 3 mm diameter probe sonicator for 30 min at 30% amplitude (Vibra-Cell Processors VCX 750; Sonics & Materials, Inc., Newtown, CT, USA). The obtained liposomes were centrifuged (Sorvall Legend X1R Centrifuge, Thermo Scientific, Waltham, MA, USA) at 15,000 rpm at 4 °C for 15 min, and the supernatant was collected and stored in a refrigerator.

To prepare methotrexate loaded-liposomes with conjugated bile acids, glycocholic acid, or taurocholic acid were weighed into a volumetric flask, mixed with 0.5 mL of methotrexate stock solution, and the volume was adjusted to 5 mL with pH 8 phosphate buffer. This prepared solution was then transferred into the test tube containing thin film, as described above. The size of the obtained liposomes was reduced using a probe sonicator, as described above.

### 2.3. Characterization of Liposomes

#### 2.3.1. Particle Size, Polydispersity Index (PDI) and Surface Charge

Using a particle size analyzer (Zetasizer Nano-ZS; Malvern Panalytical Ltd, Malvern, UK), dynamic light scattering was used to measure the average particle size, PDI, and zeta potential of various methotrexate-loaded liposome formulations. The samples were filled into a foldable capillary zeta cell and sealed with stoppers after being diluted with a suitable quantity of phosphate buffer (pH 8) prior to the measurement. Every measurement was carried out three times.

#### 2.3.2. Percent Entrapment Efficiency (%EE)

To evaluate the %EE of the drug in liposomes, the amount of methotrexate was varied between 5% and 50% of the liposomal formulation’s lipid weight (phospholipid and cholesterol). The amount of methotrexate entrapped in liposomes was determined using centrifugal filters with a molecular weight cut of 30,000 daltons (Amicon Ultra0.5; Merck KGaA, Darmstadt, Germany). Liposomes (0.5 mL) were pipetted into the sample reservoir, inserted in a retentate vial, and centrifuged at 10,000× *g* at 4 °C for 50 min. The obtained filtrate was discarded, and the sample reservoir was inserted into a new retentate vial. Liposomes were disrupted by adding 0.2 mL of 0.1% *w*/*v* Triton-X 100 (VWR, Radnor, PA, USA) and centrifuged at 10,000× *g* at 4 °C for 10 min. The obtained filtrate was analyzed for the methotrexate concentration by HPLC, and the %EE was calculated using the following equation (Equation (1)):% EE = (Am/Ai) × 100(1)
where Am is the amount of methotrexate analyzed using the method described above and Ai is the initial amount of methotrexate in the formulation.

### 2.4. In Vitro Transport Across the Blood-Brain Barrier Model

#### 2.4.1. Cell Culture

The in vitro transport of methotrexate solution and various formulations of methotrexate-loaded liposomes across the blood-brain barrier model was performed using a monoculture of human cerebral microvascular endothelial cells (HBEC-5i; ATCC, Manassas, VA, USA). HBEC-5i cells were grown in DMEM/Ham’s F-12 medium (Corning Inc., Corning, NY, USA) supplemented with 40 µg/mL of endothelial cell growth supplement (ECGS), 10% fetal bovine serum, and 1% penicillin-streptomycin; they were then incubated at 37 °C with 5% CO_2_.

#### 2.4.2. In Vitro Blood-Brain Barrier Model Construction

HBEC-5i cells were seeded into the inserts at a density of 7.5 × 10^4^ cells/well on 12 mm Transwell^®^ plates with a 3-µm pore size PET membrane insert and a 1.12-cm^2^ membrane growth area (Corning Inc., Corning, NY, USA). The cells were allowed to grow for 4 days, and the medium was changed every other day. The barrier integrity of the blood-brain barrier models was evaluated by measuring transendothelial electrical resistance (TEER) using a Millicell® ERS-2 voltohmmeter (Merck KGaA, Darmstadt, Germany). The TEER value of the insert with cells was subtracted from that of the blank insert (without cells) and reported per membrane area (Ωcm^2^), as shown in Equation (2).TEER = R(Ω) × A(cm^2^)(2)
where R is the electrical resistance of the monolayer subtracted from that of the blank insert and A is the surface area of the insert.

The tested formulation was added to the apical chamber inside the insert (0.5 mL), whereas the basolateral side was filled with 1.5 mL of medium (DMEM/Ham’s F-12 medium). At predetermined times of 15, 30, 60, 90, and 120 min, 0.3 mL of the medium was withdrawn, and the methotrexate concentration was determined by HPLC. The same volume of the medium was replaced immediately to maintain a constant volume. Each experiment was performed in triplicate. The cumulative amount of the drug that penetrated the basolateral side was plotted against time by fitting with three mathematical models, namely, the zero-order model (Equation (3)), first-order model (Equation (4)), and Higushi model (Equation (5)), as follows:A = A_0_ + K_0_t(3)lnA = lnA_0_ + K_1_t(4)(5)$$A=K_{H}+\sqrt{t}$$
where A is the cumulative amount of methotrexate at time (t), A_0_ is the initial amount of the drug, t is time (minutes), K_0_ is the zero-order penetration constant, K_1_ is the first-order penetration constant, and K_H_ is the Higushi order penetration constant.

The penetration rate was calculated from the slope of the kinetic model, which provided the highest coefficient of determination (R^2^).

### 2.5. Transport Pathways Across the Blood-Brain Barrier of Liposomes

There are many proposed transport routes across the blood-brain barrier, including the transcellular pathway, paracellular pathway, carrier-mediated transport, receptor-mediated transcytosis, adsorptive-mediated transcytosis, and cell-mediated transcytosis [19,20]. Adsorptive-mediated transcytosis or pinocytosis can be classified as macropinocytosis, clathrin-mediated endocytosis (CME), caveolae-mediated endocytosis (CVME), or clathrin and caveolae-independent endocytosis [21]. There have been no reports on the transport route of liposomes via pinocytosis of brain endothelial cells. Therefore, this study investigated the internalization pathways of liposomes using four specific endocytosis inhibitors. Chlorpromazine was used as a specific inhibitor of clathrin-mediated endocytosis by dissociating the clathrin lattice. Genistein was used as a specific inhibitor of caveolae-mediated endocytosis by inhibiting tyrosine kinase. Filipin was used as an inhibitor of caveolae-mediated endocytosis by binding to cholesterol. Amiloride was used as a specific macropinocytosis inhibitor by inhibiting Na/H exchange proteins.

#### 2.5.1. Specific Inhibitor Cytotoxicity Test

The MTT cell viability assay was used to investigate the cytotoxicity of certain inhibitors to HBEC-5i cells. Cells were seeded in 96-well plates at a density of 80,000 cells/mL in a 100 µL volume and incubated at 37 °C for 24 h. Following the removal of the medium, 100 µL of PBS was used to wash the cells. Following the addition of 100 µL of each inhibitor, the entire mixture was incubated for 24 h at 37 °C. Following the removal of the inhibitor, 100 µL of PBS was used to wash the cells. After incubation, the MTT solution was discarded and 100 µL of DMSO was added to dissolve the formazan crystals. Utilizing a microplate reader (FLUOstar Omega; BMG LABTECH, Ortenberg, Germany), the absorbance of the resultant solution was determined at a wavelength of 550 nm. By comparing it to that of untreated cells (negative control), which was considered to be 100% vitality, the cell viability percentage was determined. As a positive control, cells incubated with 0.1% *w*/*v* Triton X-100 were deemed to exhibit 100% cell death. For the cellular internalization study, the inhibitor concentration that yielded at least 80% cell viability was employed.

#### 2.5.2. Preparation of Rhodamine DHPE-Labeled Liposomes

The liposomal formulation, which had the highest penetration rate from in vitro transport across the blood-brain barrier model, was selected as a candidate for cellular internalization pathway evaluation. The liposomes were labeled with rhodamine DHPE, which exhibited red fluorescence. To prepare rhodamine DHPE-labeled liposomes, rhodamine DHPE was used at a ratio of 1:100 *w*/*w*, in contrast to the phospholipid and cholesterol used in the liposomal formulation. Rhodamine DHPE (5 mg) was dissolved in 1 mL of chloroform–methanol (2:1 *v*/*v*) as a stock solution. A total of 85 µL of the rhodamine DHPE stock solution was pipetted into a test tube containing phospholipid and cholesterol solution. Rhodamine DHPE-labeled liposomes were prepared using the same process described in Section 2.2.3 without the addition of methotrexate.

#### 2.5.3. Flow Cytometry

HBEC-5i cells were plated in 24-well plates at a density of 60,000 cells/mL for 1 mL and incubated at 37 °C for 24 h. Following incubation, the medium was removed and 1 mL of each specific inhibitor was added, allowing the cells to incubate for an additional 4 h. Untreated cells served as positive controls. The inhibitors were then discarded, and the cells were rinsed with 1 mL of PBS. Rhodamine DHPE-labeled liposomes were diluted in a 1:1 v/v ratio with medium before being introduced into each well (1 mL) and incubated for another 4 h. After incubation, the liposome solution was removed and cells were washed twice with 1 mL of PBS. The cells were detached using 200 µL of trypsin-EDTA for 3 min at 37 °C. The resulting cell suspension was combined with 500 µL of medium and centrifuged at 1500 rpm for 5 min. A total of 650 µL of the supernatant was discarded, and the pellet was resuspended in 400 µL of 4% *v*/*v* formaldehyde in PBS for a flow cytometry analysis. The analysis was performed using a CytoFlex flow cytometer (Beckman Coulter, Indianapolis, IN, USA) equipped with a 488 nm excitation laser. Fluorescence detection was conducted using a 585/42 nm bandpass filter. All experiments were performed in triplicate for quantitative analysis.

#### 2.5.4. Confocal Laser Scanning Microscopy (CLSM)

CLSM was used to visualize the cellular internalization of fluorescently labeled liposomes. HBEC-5i cells at a density of 70,000 cells/mL were seeded (1 mL) into 24-well plates, having a round coverslip, and incubated for 48 h at 37 °C. Then, the medium was removed, and 1 mL of each specific inhibitor solution was added to the well and incubated for 4 h. After that, the specific inhibitor solution was removed and the cells were washed with 1 mL of PBS.

Rhodamine DHPE-labeled liposomes were diluted with medium at a ratio of 1:1 *v*/*v* before being added to the wells (1 mL) and incubated for 4 h. Cells treated with rhodamine DHPE-labeled liposomes without the addition of a specific inhibitor were used as a positive control. Cells were not treated with both rhodamine DHPE-labeled liposomes, and a specific inhibitor solution was used as a negative control.

Alexa Fluor 488 phalloidin was used to stain the filamentous actin of cells by dissolving it in ICC buffer before cell staining. ICC buffer was prepared by mixing 0.1% *w*/*v* Triton X-100 with 5 mg/mL of bovine serum albumin before being adjusted to 50 mL using PBS. Then, Alexa Fluor 488 phalloidin were dissolved in 1.5 mL of methanol to make a concentration of 6.6 µM as a stock solution. Alexa Fluor™ 488 phalloidin solution was prepared by mixing 125 µL of Alexa Fluor 488 phalloidin stock solution with 5 mL of ICC buffer.

After treatment with rhodamine DHPE-labeled liposomes for 4 h, the liposomes were removed and the cells were washed with 1 mL of PBS 2 times. The cells were fixed with 1 mL of 3.7% *v*/*v* formaldehyde solution for 10 min. Then, the formaldehyde solution was removed and the cells were washed with 1 mL of PBS. After that, 1 mL of ICC buffer was added to the well and the cells were incubated for 30 min at room temperature. The ICC buffer was removed, and the cells were stained with 500 µL of Alexa Fluor™ 488 phalloidin solution for 30 min. The Alexa Fluor™ 488 phalloidin solution was removed, and the cells were washed with 1 mL of PBS 3 times. To stain the nucleus, 1 mL of 2 µg/mL of DAPI dihydrochloride aqueous solution was added to each well for 1 h. After that, the cells were washed with 1 mL of PBS 3 times. The coverslip was mounted on a glass slide with 50% w/w glycerin in PBS.

The stained HBEC-5i cells were observed by inverted confocal laser scanning microscopy (Zeiss LSM 800-Airy scan; Carl Zeiss, Jena, Germany) using a Plan-Apochromat 63×/1.4 oil immersion objective lens. The images were acquired using ZEN Blue Edition software version 2.3 lite (Carl Zeiss Microscopy GmbH, Jena, Germany).

### 2.6. High-Performance Liquid Chromatography (HPLC)

Methotrexate was analyzed using HPLC (Shimadzu, Kyoto, Japan) following a validated method by Oliveira et al. [22]. The separation was performed on a C18 re-versed-phase column (InertSustain C18, GL Sciences, Tokyo, Japan) with a 5-µm particle size and dimensions of 4.6 mm × 150 mm. The mobile phase consisted of 50 mM ammonium acetate buffer (pH 6) and methanol in a 75:25 *v*/*v* ratio. The buffer was prepared by dissolving 3.854 g of ammonium acetate in water and adjusted to 1 L, with pH modification occurring using glacial acetic acid. A sample analysis was conducted by injecting 25 µL of the mobile phase at a flow rate of 1 mL/min, and detection was performed using a UV detector set at 302 nm.

### 2.7. Statistical Analysis

All the data were statistically analyzed using a paired-sample *t* test and one-way analysis of variance (ANOVA) followed by a post hoc test (LSD). Values of *p* < 0.05 were considered to indicate statistically significant differences.

## 3. Results and Discussion

### 3.1. Physicochemical Properties of Liposomes

#### 3.1.1. Particle Size, Polydispersity Index (PDI) and Surface Charge

The average particle size, PDI, and zeta potential of the different methotrexate-loaded liposome formulations are shown in Table 2. Among CL and liposomes with different amounts of glycocholic acid, L0.5%G had an average particle size significantly larger than that of CL, L1%G, and L2%G. CL had an average particle size significantly greater than those of L1%G and L2%G. L2%G had an average particle size significantly larger than that of L1%G. These results indicated that adding 0.5% glycocholic acid increased the particle size of the liposomes. In contrast, adding glycocholic acid at concentrations of 1% and 2% decreased the particle size of liposomes. Among CL and liposomes with different amounts of taurocholic acid, L0.5%T had an average particle size significantly larger than that of CL, L1%T, and L2%T. CL had an average particle size significantly greater than those of L1%T and L2%T. L2%T had an average particle size significantly greater than that of L1%T. These results indicated that adding 0.5% taurocholic acid increased the particle size of the liposomes. In contrast, adding taurocholic acid at concentrations of 1% and 2% decreased the particle size of liposomes. Liposomes with glycocholic acid and liposomes with taurocholic acid had different particle sizes. L0.5%T had an average particle size significantly larger than that of L0.5%G. L1%T had an average particle size larger than that of L1%G. The average particle size did not significantly differ between L2%G and L2%T. This study revealed that adding glycocholic acid and taurocholic acid to liposomes at a concentration of 0.5% could increase the particle size. In comparison, adding glycocholic acid and taurocholic acid to liposomes at 1% and 2% decreased the particle size of the liposomes. At concentrations of 0.5% and 1%, the addition of glycocholic acid could reduce the particle size of liposomes more than the addition of taurocholic acid.

The PDI of the liposomal formulations ranged from 0.068 to 0.387, indicating that the prepared liposomes had a narrow size distribution. The zeta potentials of all methotrexate-loaded liposomal formulations were negative (−1.43 to −19.23 mV). The phospholipid used in this study was phosphatidylcholine, a zwitterionic surfactant with a mean isoelectric point of 6.7 [23]. The methotrexate structure also contains dicarboxylic acid, with pKa values of 4.8 and 5.5. In this study, in phosphate buffer at pH 8, the liposome particles exhibited negative charges. In addition, adding glycocholic acid and taurocholic acid, which are carboxylic acids, to the formulation increased the negative charge of the liposomes.

#### 3.1.2. % EE

The effect of methotrexate amounts varying from 5% to 50% of the lipid weight on the entrapment efficiency of conventional liposomes is shown in Figure 1. An increase in the amount of methotrexate to 20% led to an increase in the % EE, with the highest EE reaching 41.14 ± 1.83. However, when the amount of methotrexate increased above 20%, the % EE gradually decreased. Thus, 20% methotrexate per lipid weight (0.17% *w*/*v* of liposome formulation) was chosen for further study.

### 3.2. In Vitro Transport Across the Blood-Brain Barrier Model

The penetration kinetics and penetration rate of the methotrexate solution as a control and each methotrexate-loaded liposome formulation are shown in Table 3. All formulations provided the best fit with the Higushi kinetic model. L1%G had a significantly greater penetration rate than the solution, CL, L0.5%G, L0.5%T, L1%T, and L2%T. There was no significant difference between L1%G and L2%G. CL had a significantly greater penetration rate than the solution. L0.5%G had a greater penetration rate than the solution and L0.5%T. There was no significant difference between L0.5% G and CL. L2%G had a significantly greater penetration rate than the solution and L0.5%T. There were no significant differences among L2%G, CL, L0.5%G, L1%G, L1%T and L2%T. Regarding the above results, the average particle size of L1%G (30.21 nm) was significantly less than L0.5%G (120.73 nm). Thus, the smaller particle size could penetrate across the blood-brain barrier better than the larger size.

There were no significant differences among L0.5%T, solution, CL, and L1%T. L1%T had a significantly greater penetration rate than solution. There were no significant differences among L1%T, CL, L0.5%G, L2%G, L0.5%T, and L2%T. L2%T had a significantly greater penetration rate than solution and L0.5%T. Among liposome formulations, L0.5%T had the lowest penetration rate of 45.43 but exhibited the largest particle size (147.87 nm). Therefore, not only the type of bile acid composition but also the particle size which affected the penetration rate across the blood-brain barrier.

In summary, the addition of glycocholic acid to liposomes at a concentration of 1% could increase the penetration rate of methotrexate across the blood-brain barrier more than methotrexate solution or conventional liposomes. In contrast, adding taurocholic acid to liposomes did not increase the penetration of methotrexate across the blood-brain barrier. Liposomes with 1% glycocholic acid had the highest penetration rate; therefore, they were selected to investigate the transport pathways across the blood-brain barrier.

The semipermeable membrane property of the blood-brain barrier results from brain microvascular endothelial cells supported by pericytes, astrocytes, and neurons. In addition, tight junctions, which are present between adjacent endothelial cells, also play an important role by limiting the passive diffusion of molecules. Tight junctions are located at the apical side of endothelial cells and are formed by essential transmembrane proteins, junctional adhesion molecules, claudins, and occludin [24]. Thus, tight junctions are integral to limiting the penetration of water-soluble molecules via the paracellular pathway. The integrity of the tight junction could be determined from the TEER value. Table 4 shows each formulation’s TEER value before and after in vitro blood-brain barrier transport. There was no significant difference between the TEER value before and after the in vitro blood-brain barrier transport of all formulations. This indicated that methotrexate, glycocholic acid and taurocholic acid did not affect the integrity of tight junctions. This study suggested that liposomes with 1% glycocholic acid, which provided the highest penetration rate, did not use the paracellular pathway as a major pathway for enhancing their ability to penetrate the blood-brain barrier.

### 3.3. Transport Pathways Across the Blood-Brain Barrier of Liposomes

Figure 2 shows the in vitro cytotoxicity of each endocytosis inhibitor on the viability of HBEC-5i cells. The concentrations of chlorpromazine, genistein, filipin, and amiloride used to study the uptake of liposomes via pinocytosis by flow cytometry and confocal laser scanning microscopy were 10 µM, 70 µM, 20 µM, and 100 µM, respectively. Liposomes with 1% glycocholic acid had the highest penetration rate of methotrexate; this formulation was selected for transportation pathway evaluation.

The fluorescence histograms obtained by flow cytometry of the control and all the specific inhibitors are shown in Figure 3. The percentage of cellular uptake of rhodamine DHPE-labeled liposomes with 1% glycocholic acid as a control and endocytosis inhibitor was 30.24–36.28%, as shown in Table 5. There was no significant difference among the control and cells treated with each specific inhibitor.

CLSM images of the negative and positive controls are shown in Figure 4a,b, respectively. There was an absence of red fluorescence for the negative control, as shown in Figure 4(a-1), whereas red fluorescence was clearly presented for the positive control, as shown in Figure 4(b-1). CLSM images of cells treated with chlorpromazine, genistein, filipin, and amiloride are shown in Figure 5a–d, respectively. Red fluorescence from the rhodamine DHPE-labeled liposomes appeared for all endocytosis inhibitors, as shown in Figure 5(a-1)–(d-1).

Theoretically, nanoparticles can be transported across the blood-brain barrier via paracellular and transcellular pathways [25]. In this study, the TEER values before and after treatment with liposomes containing 1% glycocholic acid did not significantly differ. It is suggested that the transport of liposomes through the paracellular pathway might be a minor pathway. In transcellular pathways, nanoparticles can move into cells through endocytosis and nonendocytosis pathways. Endocytosis is an energy-dependent process classified as phagocytosis or pinocytosis [21]. The nonendocytosis pathway is an energy-independent process that involves the direct penetration of particles into cells [26,27]. Pinocytosis is the uptake of smaller particles than phagocytosis into cells and can be divided into macropinocytosis and receptor-mediated pinocytosis. There are several types of receptor-mediated endocytosis, such as clathrin-mediated endocytosis (CME), caveolae-mediated endocytosis (CVME), and clathrin- and caveolae-independent endocytosis. Chlorpromazine, genistein, filipin, and amiloride are specific endocytosis inhibitors used to evaluate nanoparticle macropinocytosis and receptor-mediated endocytosis. In this study, HBEC-5i cells were treated with these inhibitors. However, fluorescently labeled liposomes with 1% glycocholic acid could be observed inside the cells, indicating that the uptake of this type of liposome did not involve pinocytosis. Phagocytosis is the uptake of large particles, which generally have sizes larger than 500 nm [28]. Generally, the delivery of nanoparticles into cells by direct penetration should have a particle size of less than 50 nm [27]. The average particle size of liposomes with 1% glycocholic acid was 30.21 nm. Thus, phagocytosis was not involved in the uptake process of these liposomes. Therefore, the transport route across the blood-brain barrier of liposomes with 1% glycocholic acid was direct penetration into cells.

## 4. Conclusions

Liposomes prepared with glycocholic acid could increase the blood-brain barrier penetration of methotrexate, while taurocholic acid could not. A suitable concentration of glycocholic acid added to liposomes, which could enhance methotrexate across the blood-brain barrier, was 1%. Therefore, liposomes with 1% glycocholic acid were found to be promising candidates for further investigation in in vivo studies. This study revealed that transport pathways across the blood-brain barrier of liposomes with 1% glycocholic acid were transcellular pathways, with direct penetration into cells acting as the major penetration pathway. In contrast, the paracellular pathway was a minor pathway.

## Figures and Tables

**Figure 1 pharmaceutics-17-00269-f001:**
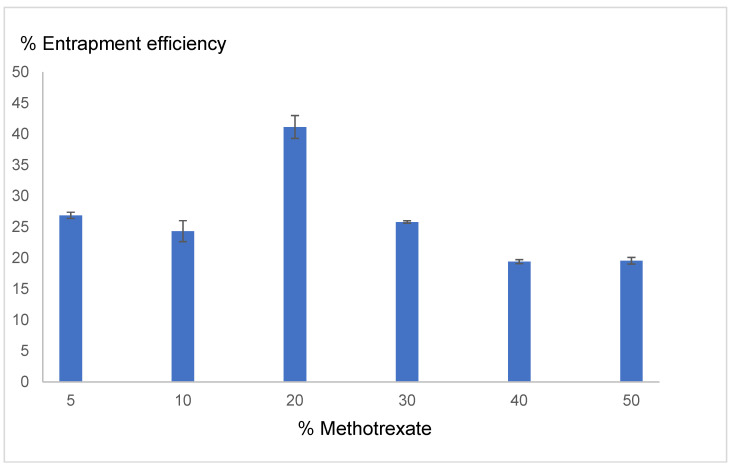
Effect of the methotrexate concentration (% per lipid weight) on the % entrapment efficiency (■). Each value represents the mean ± standard deviation (*n* = 3).

**Figure 2 pharmaceutics-17-00269-f002:**
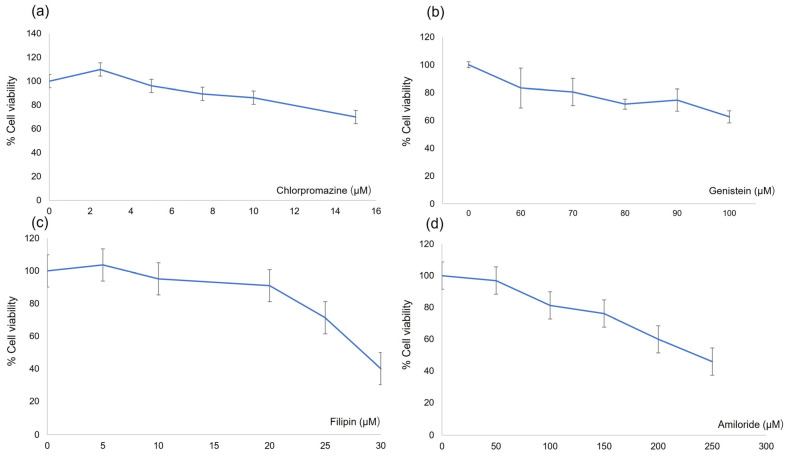
In vitro cytotoxicity test of endocytosis inhibitors. (**a**) chlorpromazine, (**b**) genistein, (**c**) filipin and (**d**) amiloride. Each value represents the mean ± standard deviation (*n* = 3).

**Figure 3 pharmaceutics-17-00269-f003:**
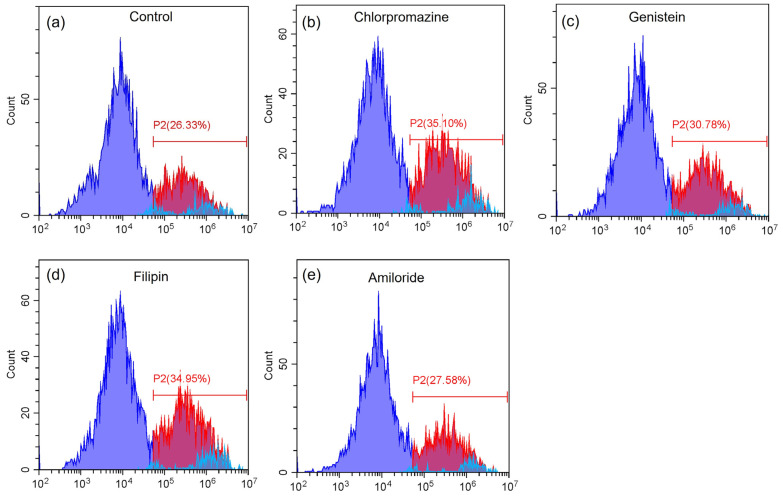
Fluorescence intensity histograms of (**a**) the control, (**b**) chlorpromazine, (**c**) genistein, (**d**) filipin and (**e**) amiloride. Purple area represents cell population without liposome-rhodamine uptake, red area represents cell population with liposome-rhodamine uptake and blue area represents auto fluorescence, respectively.

**Figure 4 pharmaceutics-17-00269-f004:**
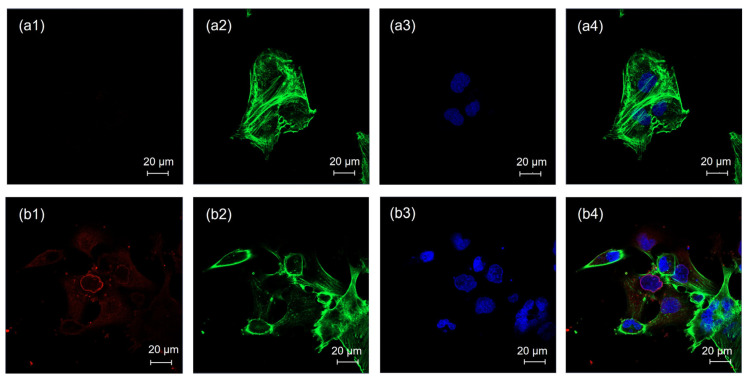
CLSM images of HBEC-5i cells treated with (**a**) the negative control and (**b**) the positive control; (**1**) the red fluorescence of rhodamine DHPE, (**2**) the green fluorescence of Alexa Fluor™ 488 phalloidin, (**3**) the blue fluorescence of DAPI, and (**4**) merged images. Each scale bar represents 20 µm.

**Figure 5 pharmaceutics-17-00269-f005:**
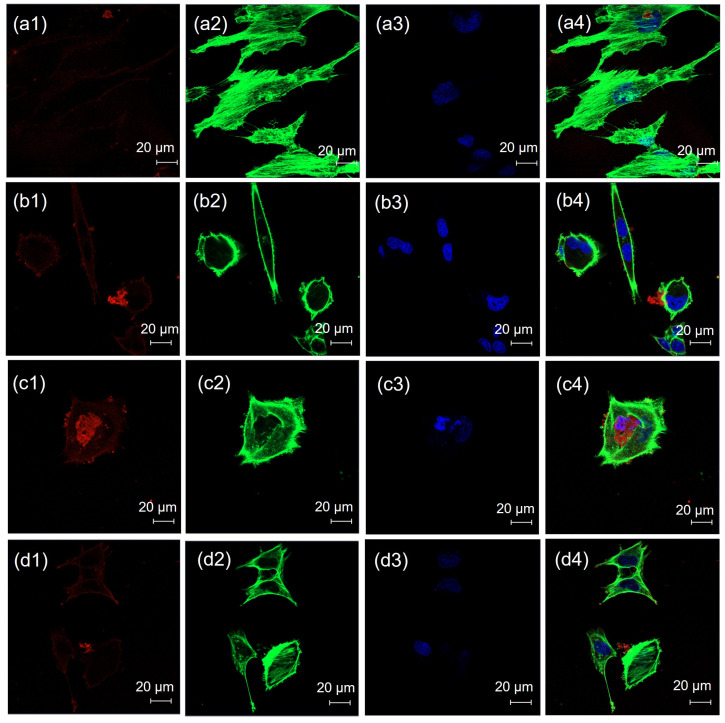
CLSM images of HBEC-5i cells treated with (**a**) chlorpromazine, (**b**) genistein, (**c**) filipin and (**d**) amiloride; (**1**) red fluorescence of rhodamine DHPE; (**2**) green fluorescence of Alexa Fluor™ 488 phalloidin; (**3**) blue fluorescence of DAPI; and (**4**) merged images. Each scale bar represents 20 µm.

**Table 1 pharmaceutics-17-00269-t001:** Composition of each liposomal formulation.

Formulation	Methotrexate (%*w*/*v*)	Phospholipid (%*w*/*v*)	Cholesterol (%*w*/*v*)	Glycocholic Acid (%*w*/*v*)	Taurocholic Acid (%*w*/*v*)	PBS pH 8 (mL)
CL	0.17	0.773	0.0773	-	-	qs 100
L0.5%G	0.17	0.773	0.0773	0.5	-	qs 100
L1%G	0.17	0.773	0.0773	1	-	qs 100
L2%G	0.17	0.773	0.0773	2	-	qs 100
L0.5%T	0.17	0.773	0.0773	-	0.5	qs 100
L1%T	0.17	0.773	0.0773	-	1	qs 100
L2%T	0.17	0.773	0.0773	-	2	qs 100

CL = Conventional Liposomes, L = Liposomes, G = Glycocholic acid, T = Taurocholic acid.

**Table 2 pharmaceutics-17-00269-t002:** Physicochemical parameters of various liposomal formulations.

Formulation	Particle Size (nm)	PDI	Zeta Potential (mV)
CL	67.48 ± 0.34	0.283 ± 0.005	−1.43 ± 0.21
L0.5%G	120.73 ± 0.81	0.195 ± 0.012	−9.85 ± 0.62
L1%G	30.21 ± 0.47	0.285 ± 0.005	−16.47 ± 0.75
L2%G	59.77 ± 0.75	0.114 ± 0.005	−19.23 ± 0.49
L0.5%T	147.87 ± 0.41	0.158 ± 0.011	−11.43 ± 0.41
L1%T	38.25 ± 1.14	0.387 ± 0.016	−11.53 ± 0.68
L2%T	60.21 ± 0.18	0.068 ± 0.008	−17.57 ± 1.37

Each value represents the mean ± standard deviation (*n* = 3).

**Table 3 pharmaceutics-17-00269-t003:** In vitro blood-brain barrier penetration kinetics and penetration rate.

Formulation	Zero Order R^2^	First Order R^2^	Higushi R^2^	$\mathrm{Higushi}\;\mathrm{Penetration}\;\mathrm{Rate}\;(\mu g/\sqrt[{2}]{h})$
Solution	0.9472 ± 0.0187	0.8671 ± 0.0941	0.9533 ± 0.0095	42.57 ± 4.17
CL	0.9748 ± 0.0116	0.8479 ± 0.0684	0.9912 ± 0.0032	52.98 ± 1.91
L0.5%G	0.9002 ± 0.0464	0.7276 ± 0.0778	0.9498 ± 0.0391	54.36 ± 0.66
L1%G	0.9743 ± 0.0178	0.8193 ± 0.0518	0.9921 ± 0.0101	64.19 ± 6.94
L2%G	0.9355 ± 0.0288	0.7078 ± 0.0478	0.9591 ± 0.0292	57.61 ± 2.83
L0.5%T	0.9609 ± 0.0329	0.8145 ± 0.0361	0.9828 ± 0.0114	45.43 ± 4.65
L1%T	0.8574 ± 0.0837	0.7607 ± 0.1035	0.9513 ± 0.0461	52.69 ± 0.99
L2%T	0.9244 ± 0.0179	0.9158 ± 0.0456	0.9811 ± 0.0221	54.22 ± 5.76

Each value represents the mean ± standard deviation (*n* = 3).

**Table 4 pharmaceutics-17-00269-t004:** TEER (Ωcm^2^) before and after in vitro blood-brain barrier transport.

Formulation	Before	After
Solution	37.78 ± 14.47	36.92 ± 0.06
CL	43.87 ± 2.38	41.07 ± 1.06
L0.5%G	39.51 ± 2.41	45.05 ± 6.25
L1% G	46.42 ± 5.59	45.31 ± 2.04
L2% G	45.33 ± 5.18	40.19 ± 10.89
L0.5%T	41.32 ± 5.84	44.81 ± 1.87
L1%T	46.29 ± 7.16	44.18 ± 7.38
L2%T	46.29 ± 5.29	41.44 ± 1.71

Each value represents the mean ± standard deviation (*n* = 3).

**Table 5 pharmaceutics-17-00269-t005:** Percentage cellular uptake from flow cytometry analysis.

Endocytosis Inhibitor	% Cellular Uptake
Control (without inhibitor)	30.24 ± 3.38
Chlorpromazine	36.28 ± 2.41
Genistein	33.04 ± 2.45
Filipin	33.12 ± 2.98
Amiloride	32.93 ± 4.64

Each value represents the mean ± standard deviation (*n* = 3).

## Data Availability

The data that support the findings of this study are available from the corresponding author upon reasonable request.

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
