# Peer review of "Development and Transportation Pathway Evaluation of Liposomes with Bile Acids for Enhancing the Blood-Brain Barrier Penetration of Methotrexate"

_pharmaceutics, 2025, doi:10.3390/pharmaceutics17020269_

Round 1
Reviewer 1 Report
Comments and Suggestions for Authors
This study aims to develop liposomes containing bile acids to enhance the penetration of methotrexate across the blood-brain barrier and to evaluate the transport pathways of the liposomes across the blood-brain barrier. The conclusion of the article states that liposomes with 1% glycocholic acid penetrate the blood-brain barrier via a transcellular pathway, entering cells directly. The term "directly entering cells" is confusing; however, the process of how this occurs is not clarified. Specific comments are as follows:
-
How were the in vitro blood-brain barrier permeability kinetics and permeability data obtained?
-
The transport pathway of liposomes with 1% glycocholic acid across the blood-brain barrier is described as direct cell penetration. How should we understand or describe this direct penetration? Are there any literature reports on similar pathways, and what are the similarities?
-
What is the mechanism of direct penetration?
-
What issues are to be clarified by the flow cytometry experiments?
Reviewer 2 Report
Comments and Suggestions for Authors
The manuscript pharmaceutics-3455640 “Development and transportation pathway evaluation of liposomes with bile acids for enhancing the blood‒brain barrier penetration of methotrexate” by Natthan Charernsriwilaiwat et al. describes the development of a nanoformulation with bile acids in the form of liposomes to enhance the penetration of methotrexate across the blood-brain barrier.
The topic of research is relevant. In the paper, modern methods of analysis were used and the references are relevant. Nevertheless, the article requires major revision before acceptance for publication.
Comments and questions:
1) Abstract: The average particle size of 30.21-120.73 nm is better written as 30 - 121 nm.
2) What is the meaning of a methotrexate concentration of 8 g/m2? What area is taken into account?
3) The abbreviation DAPI should be explained.
4) Section 2.2.1 - Why did you choose a buffer solution with a pH of 8?
5) Section 2.2.1 - What concentration of phosphoric acid was used?
6) Section 2.2.1 - It is more correctly written 1M NaOH.
7) Section 2.2.1 - What degree of purity water was used to prepare the buffer?
8) Section 2.2.3 - For the evaporator instrument, vortex mixer, and centrifuge, the manufacturers, city, and country of manufacture should be specified.
9) Section 2.2.3 - The preparation of liposomes containing bile acids is not fully described, and as a result it is not clear how they were obtained.
10) Table 1 - What is the meaning of CL abbreviation?
11) Section 2.3.2 - Liposomes may disrupt during centrifugation and therefore methotrexate may be released into the supernatant and affect the experimental data.
12) Section 2.4.2 - Which test substance solvent (apical chamber) and which release medium (basolateral chamber) did you use in this experiment?
13) Table 1 - Why does the addition of small amounts of bile acids to liposomes increase their hydrodynamic sizes almost 2-fold?
14) Section 3.1.2 - Why were only conventional liposomes chosen to influence the amount of loaded methotrexate? It is of great interest to study the effect of bile acid additives on the entrapment efficiency of methotrexate.
15) Section 3.2 - Results of permeability studies of methotrexate-containing liposomes on the in vitro transport across the blood-brain barrier model first of all should be represented in the form of a permeability coefficient. Since a comparison of the permeability coefficients of different formulations allows us to evaluate their promising potential. It is not clear from Table 1 how the penetration rate of liposome-loaded methotrexate differs from that of the pure drug. In the comments to Table 1, it would be good to indicate the statistical data supporting the difference between the obtained values.
16) Table 5 - Are there statistically significant differences between the values of the cellular uptake.
17) It would be good to add an outlook of this study in the Conclusions.
18) In addition, the English grammar should be improved.
Reviewer 3 Report
Comments and Suggestions for Authors
This manuscript is an interesting work about development liposomes with bile acids to enhance the blood-brain barrier penetration of methotrexate, which is certainly a relevant topic. It should be noted that the presented study is important in view of the fact that, evaluate the transport pathway across the blood-brain barrier of liposomes remains a challenge, necessitating the researching.
- It is known that the size of liposomes affects their penetration through the blood-brain barrier. The authors should comment on this and add information in the introduction on how liposomes of different sizes affect the rate and kinetics of penetration.
- For examples, L1%G had a significantly greater penetration rate (64.19±6.94) than the solution, CL, L0.5%G, L0.5%T, L1%T, and L2%T. But the L1%G sample had the smallest liposome size (30.21±0.47). Moreover, the lowest penetration rate (45.43±4.65) is observed in the L0.5%T sample, which has the largest liposome size (147.87±0.41). Perhaps, in this case, the high penetration rate is associated not only with the composition of the liposomes, but also with their small size, and vice versa, the low permeation rate of liposomes across the blood-brain barrier is associated with the large size of the liposomes? The authors need to answer this question.
The results here are certainly of sufficient interest to deserve publication in Journal Pharmaceuticals. Thus, this manuscript can be accepted with a major revision once the comments have been corrected.
Round 2
Reviewer 1 Report
Comments and Suggestions for Authors
The authors have addressed my concerns in a comprehensive manner, which has significantly improved the quality of the manuscript.
Reviewer 2 Report
Comments and Suggestions for Authors
The manuscript has been revised and can be accepted.
However, I recommend correcting the particle sizes throughout the text from 30.21 nm and 120.73 nm to 30 nm and 121 nm, and 45.43 and 147.87 to 45 and 145 etc. (e.g., on pages 17, 18, and all particle sizes in the Table 2), because the dynamic light scattering method does not have such high accuracy.
Reviewer 3 Report
Comments and Suggestions for Authors
The manuscript sufficiently improved and can be accepted in Journal Pharmaceuticals.